# Grassmannian Shape Representations for Aerodynamic Applications

**Olga A. Doronina,**[1] **Zachary J. Grey,** [2] **Andrew Glaws** [1]

[1] National Renewable Energy Laboratory, Golden, CO, USA
[2] National Institute of Standards and Technology, Boulder, CO, USA
olga.doronina@nrel.gov, zachary.grey@nist.gov, andrew.glaws@nrel.gov

## Abstract

Airfoil shape design is a classical problem in engineering and manufacturing. Our motivation is to combine principled physics-based considerations for the shape design problem with modern computational techniques informed by a data-driven approach. Traditional analyses of airfoil shapes emphasize a flow-based sensitivity to deformations which can be represented generally by affine transformations (rotation, scaling, shearing, translation). We present a novel representation of shapes which decouples affine-style deformations from a rich set of data-driven deformations over a submanifold of the Grassmannian. The Grassmannian representation, informed by a database of physically relevant airfoils, offers (i) a rich set of novel 2D airfoil deformations not previously captured in the data, (ii) improved low-dimensional parameter domain for inferential statistics informing design/manufacturing, and (iii) consistent 3D blade representation and perturbation over a sequence of nominal shapes.

## Introduction

Many AI-aided design and manufacturing algorithms rely on shape parametrization methods to manipulate shapes in order to study sensitivities, approximate inverse problems, and inform optimizations. Two-dimensional cross-sections of aerodynamic structures such as aircraft wings or wind turbine blades, also known as airfoils, are critical engineering shapes whose design and manufacturing can have significant impacts on the aerospace and energy industries. Research into AI and ML algorithms involving airfoil design for improved aerodynamic, structural, and acoustic performance is a rapidly growing area of work (Zhang, Sung, and Mavris 2018; Li, Bouhlel, and Martins 2019; Chen, Chiu, and Fuge 2019; Glaws et al. 2021; Jing et al. 2021; Yonekura and Suzuki 2021; Yang, Lee, and Yee 2021).

While airfoil shapes can appear relatively benign, their representation and design are complex due to their extreme operating conditions in use and the highly sensitive relationship between deformations to the shape and changes in aerodynamic performance. The current state-of-the-art for airfoil shape parametrization is the class-shape transformation (CST) method (Kulfan 2008). In this approach, the upper and lower surfaces of an airfoil are each defined using a class function to set the general class of the geometry to an airfoil, and a shape function that usually takes the form of a Bernstein polynomial expansion to describe a specific shape. The coefficients in this polynomial expansion are typically treated as tuning parameters to define new airfoil shapes. However, defining a meaningful design space of CST parameters across a collection of airfoil types is difficult. That is, it is challenging to interpret how modified CST parameters will perturb the shape and thus difficult to contain or bound CST parameters to produce "reasonable" aerodynamic shapes. Furthermore, CST representations couple large-scale affine-type deformations—deformations resulting in significant and relatively well-understood impacts to aerodynamic performance—with undulating perturbations that are of increasing interest to airfoil designers across industries. This coupling between physically meaningful affine deformations and undulations in shapes resulting from higher-order polynomial perturbations complicates the design process.

In this work, we explore a data-driven approach that uses a Grassmannian framework to represent airfoil shapes. The resulting set of deformations to airfoil shapes is independent of the very important and often constrained affine deformations. Modern airfoil design often incorporates constrained design characteristics of twist (or angle-of-attack) and scale which must be fixed or treated independently of higher-order deformations to a shape such as a rich set of changing inflections. Our approach decouples these two aspects of airfoil design and offers new interpretations of a space of shapes, not previously considered. In what follows, we provide a brief overview of the airfoil representation scheme and demonstrate its flexibility over current methods, including the capability to extend from two-dimensional airfoils to full three-dimensional wind turbine blades.

## Discrete representation & deformation

In general, a shape can be represented as a boundary defined by the closed (injective) curve $\boldsymbol{c} : \mathcal{I} \subset \mathbb{R} \to \mathbb{R}^2 : s \mapsto \boldsymbol{c}(s)$ over a compact domain $\mathcal{I}$ which can be arbitrarily reparametrized to $[0, 1]$. In practice, we represent the 2D airfoil shape as an ordered sequence of $n$ *landmarks* $(\boldsymbol{x}_i) \in \mathbb{R}^2$ for $i = 1, \ldots, n$. That is, given some curve $\boldsymbol{c}(s)$, we have landmark points $\boldsymbol{x}_i = \boldsymbol{c}(s_i)$ for $0 \leq s_1 < s_2 < \cdots < s_n \leq 1$. Moving along the curve, this sequence of planar vectors defining the airfoil shape results in the matrix $\boldsymbol{X} = [\boldsymbol{x}_1, \ldots, \boldsymbol{x}_n]^\top \in \mathbb{R}_*^{n \times 2}$, where $\mathbb{R}_*^{n \times 2}$ refers to the

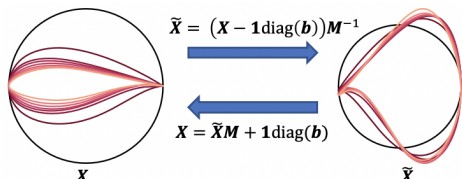

$$\tilde{\boldsymbol{X}} = (\boldsymbol{X} - \mathbf{1}\mathrm{diag}(\boldsymbol{b}))\boldsymbol{M}^{-1}$$

$$\boldsymbol{X} = \tilde{\boldsymbol{X}}\boldsymbol{M} + \mathbf{1}\mathrm{diag}(\boldsymbol{b})$$

$\boldsymbol{X}$   $\tilde{\boldsymbol{X}}$

Figure 1: Collection of cross-sectional airfoils defining IEA 15MW blade in physical (left) and Landmark-Affine standardized coordinates (right).

space of full-rank $n \times 2$ matrices. This full-rank restriction ensures that we do not consider degenerate $\boldsymbol{X}$ as a feasible discrete representation of an airfoil shape.

The innovative characteristic of the proposed approach is representing airfoil shapes as elements of a Grassmann manifold (Grassmannian) $\mathcal{G}(n, 2)$ paired with a corresponding affine transformation (invertible 2-by-2 matrices and translation) representing a subset of rotation, scaling, and shearing shape deformations. This definition of the airfoil shape makes important subsets of deformations independent, allowing designers to make interpretable and systematic changes to airfoil shapes. For example, one may seek to preserve the average airfoil thickness or camber while independently studying all remaining deformations as perturbations over the Grassmannian.

### Affine deformations

Affine deformations of an airfoil have the form $\boldsymbol{M}^\top \boldsymbol{c}(s) + \boldsymbol{b}$, where $\boldsymbol{M} \in GL_2$ is an element from the set of all invertible $2 \times 2$ matrices[1] and $\boldsymbol{b} \in \mathbb{R}^2$. For a discrete shape representation, affine deformations can be written as the smooth right action with translation $\boldsymbol{X}\boldsymbol{M} + \mathbf{1}\mathrm{diag}(\boldsymbol{b})$, where $\mathbf{1}$ denotes the $n$-by-2 matrix of ones. The translation of the shape $\boldsymbol{b}$ does not change the intrinsic characteristics of the shape (i.e., it has no deforming effect) and is generally of little interest if not to locate shapes relative to one another (e.g., in 3D blade design) or to define a center of rotation. Focusing on the linear term $\boldsymbol{M}$, we can identify four types of physically meaningful deformations as one-parameter subgroups through $GL_2$: (i) changes in thickness, (ii) changes in camber, (iii) changes in chord, and (iv) changes in twist (rotation or angle-of-attack) or some composition thereof. These deformations can be represented by specific forms $\boldsymbol{M}_t$ with $t \in (0, 1)$, respectively, as

$$\text{(i) } \boldsymbol{M}_t \triangleq \begin{bmatrix} 1 & 0 \\ 0 & t \end{bmatrix}, \quad \text{(ii) } \boldsymbol{M}_t \triangleq 2\begin{bmatrix} (1-t) & 0 \\ 0 & t, \end{bmatrix},$$

$$\text{(iii) } \boldsymbol{M}_t \triangleq \begin{bmatrix} t & 0 \\ 0 & 1 \end{bmatrix}, \quad \text{(iv) } \boldsymbol{M}_t \triangleq \begin{bmatrix} \cos(\frac{t\pi}{2}) & -\sin(\frac{t\pi}{2}) \\ \sin(\frac{t\pi}{2}) & \cos(\frac{t\pi}{2}) \end{bmatrix}.$$

Sensitivity analysis involving CST parameters (Grey and Constantine 2018) has revealed certain shape deformations that change transonic coefficients of lift and drag the most, on average, are very similar to physical deformations of

the form (i) and (ii)—a result that resonates with laminar flow theory. The dominating impact of these perturbations on aerodynamic quantities of interest inhibits the study of a richer set of perturbations to airfoil shapes. Note that a set of "dents" and "dings" (changing inflection) common to damage and manufacturing defects in an airfoil shape are not well described by affine deformations. This motivates the need for a set of parameters describing deformations independent of those in the dominating class of affine transformations (more precisely, transformations as smooth right actions over $GL_2$). This line of research was initially proposed as an extension of (Grey and Constantine 2018) in (Grey 2019).

Although the presented affine deformations only constitute a subset of important aerodynamic deformations over $GL_2$, we contend that aerodynamic quantities will be significantly influenced by any other combination, composition, or generalization of the presented affine deformations so long as they remain elements in $GL_2$—deformations by rank deficient $\boldsymbol{M}$, which collapse landmarks to a line or the origin, are not considered physically relevant. These affine deformations are important for design and are usually constrained or rigorously chosen when selecting nominal definitions of shapes for subsequent numerical studies and 3D blade definition. We seek to decouple and preserve these features through a set of inferred deformations over the Grassmannian that are independent of $GL_2$.

### Grassmannian representation

The Grassmannian[2] $\mathcal{G}(n, q)$ is the space of all $q$-dimensional subspaces of $\mathbb{R}^n$. Note that for (planar) airfoil design, we consider $q = 2$. Formally, $\mathcal{G}(n, q) \cong \mathbb{R}_*^{n \times q}/GL_q$ and $\tilde{\boldsymbol{X}} \in \mathbb{R}_*^{n \times q}$ is a full-rank representative element of an equivalence class $[\tilde{\boldsymbol{X}}] \in \mathcal{G}(n, q)$ of all matrices with equivalent span (Absil, Mahony, and Sepulchre 2008). In this way, every element of the Grassmannian is a full-rank matrix modulo $GL_q$ deformations, and elements of the Grassmannian are (by definition) decoupled from the aerodynamically important affine deformations (e.g., variations in camber or thickness) discussed in the previous section. This enables deformations over $\mathcal{G}(n, q)$ that are independent of affine deformations. Furthermore, we can sample a data-driven submanifold of $\mathcal{G}(n, q)$ preserving these important affine transformations or parametrizing them independently.

It is common (Edelman, Arias, and Smith 1998; Gallivan et al. 2003) to view the Grassmannian as a quotient topology of orthogonal subgroups such that $\tilde{\boldsymbol{X}}^\top \tilde{\boldsymbol{X}} = \boldsymbol{I}_q$—i.e., the $n$ landmarks in $\mathbb{R}^q$ have sample covariance proportional to the $q \times q$ identity $\boldsymbol{I}_q$. Therefore, a representative computational element of the Grassmannian is an $n \times q$ matrix with orthonormal columns (Edelman, Arias, and Smith 1998).[3] This offers certain computational advantages and motivates a scaling of airfoil landmark data for computations over $\mathcal{G}(n, 2)$ for airfoil design (Bryner et al. 2014; Grey 2019).

---

[1]For brevity, we simply refer to $GL_2(\mathbb{R})$ as $GL_2$ since all data and computation is over the reals.

[2]We assume the Riemannian metric $\mathrm{tr}(\boldsymbol{A}^\top \boldsymbol{B})$ inherited from embedding space (Absil, Mahony, and Sepulchre 2008).

[3]In our case, $n$ is equal to the number of landmarks and $q = 2$ is the dimension of the ambient space where the shape lives.

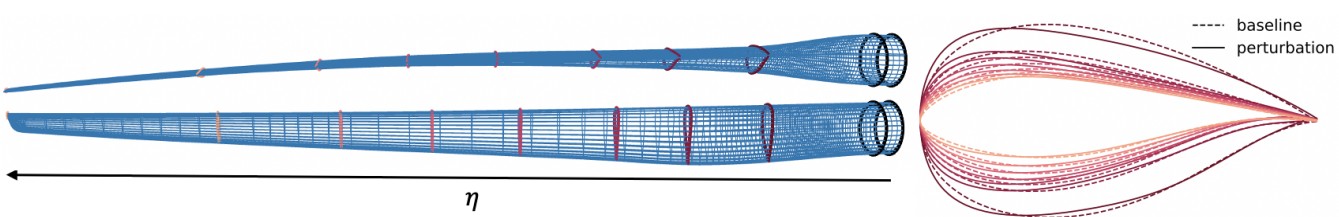

Figure 2: Example of a wire frame of a perturbed IEA-15MW blade obtained from interpolation of the solid-color cross-sections. Note that consistent perturbations to the shape are applied to all of the baseline airfoils in the blade.

To represent physical airfoil shapes as elements of the Grassmannian, we apply Landmark-Affine (LA) standardization (Bryner et al. 2014). LA-standardization normalizes the shape such that it has zero mean (without loss of generality) and sample covariance proportional to $I_2$ over the $n$ discrete boundary landmarks defining the shape. Given an airfoil shape $X \in \mathbb{R}_*^{n \times 2}$, let $M$ be the 2-by-2 invertible matrix computed via the thin singular value decomposition (SVD) of $X^\top$ and $b \in \mathbb{R}^2$ is the two-dimensional center of mass of $X$. Then, the mapping between discrete airfoil $X$ and the paired LA-standardized representation (denoted by $\tilde{X}$) is yet another affine transformation, $X = \tilde{X} M + \mathbf{1}\mathrm{diag}(b)$. Recall that $[\tilde{X}] \in \mathcal{G}(n, 2)$ and $\tilde{X}$ is merely a representative element of the Grassmannian defined uniquely up to any $GL_2$ deformations. Figure 1 shows the transformation between these two representations.

## Grassmannian blade interpolation

The Grassmannian framework for airfoil representation has the additional benefit of enabling the design of three-dimensional wings and blades. In the context of wind energy, full blade designs are often characterized by an ordered set of planar airfoils at different blade-span positions from hub to tip of the blade as well as profiles of twist, chord scaling, and translation. Current approaches to blade design require significant hand-tuning of airfoils to ensure the construction of valid blade geometries without dimples or kinks. Our proposed approach enables the flexible design of new blades by applying consistent deformations to all airfoils and smooth interpolation of shapes between landmarks.

The mapping from airfoils to blades amounts to a smoothly varying set of affine deformations over discrete blade-span positions—a common convention in next-generation wind turbine blade design. The discrete blade can be represented as a sequence of matrices $(X_k) \in \mathbb{R}_*^{n \times 2}$ for $k = 1, \ldots, N$. However, the challenge is to interpolate these shapes from potentially distinct airfoil classes to build a refined 3D shape such that the interpolation preserves the desired affine deformations along the blade (chordal scaling composed with twist over changing pitch axis).

Given an induced sequence of equivalence classes $([\tilde{X}_k]) \in \mathcal{G}(n, 2)$ for $k = 1, ..., N$ at discrete blade-span positions $\eta_k \in \mathcal{S} \subset \mathbb{R}$ from a given blade definition (see the colored curves in Figure 2), we can construct a piecewise geodesic path over the Grassmannian to interpolate discrete blade shapes independent of affine deformations.

That is, we utilize a mapping $\tilde{\gamma}_{k,k+1} : [\tilde{X}_k] \mapsto [\tilde{X}_{k+1}]$ as the geodesic interpolating from one representative LA-standardized shape to the next (Edelman, Arias, and Smith 1998).[4] Thus, a full blade shape can be defined by interpolating LA-standardized airfoil shapes using these piecewise-geodesics over ordered blade-span positions $\eta_k$ along a non-linear representative manifold of shapes. Finally, to get interpolated shapes back into physically relevant scales, we apply inverse affine transformation based on previously constructed splines defining the carefully designed affine deformations,

$$X(\eta) = \tilde{X}(\eta) M(\eta) + \mathbf{1}\mathrm{diag}(b(\eta)). \tag{1}$$

An important caveat when inverting the shapes in (1) back to the physically relevant scales for subsequent twist and chordal deformations is a *Procrustes clustering*. From the blade tip shape $\tilde{X}_N$ to the blade hub shape $\tilde{X}_1$, we sequentially match the representative LA-standardized shapes via Procrustes analysis (Gower 1975). This offers rotations that can be applied to representative LA-standardized airfoils for matching—which do not fundamentally modify the elements in the Grassmannian. Consequently, we cluster the sequence of representative shapes $\tilde{X}_k$ by optimal rotations in each $[\tilde{X}_k]$ to ensure they are best oriented from tip to hub to mitigate concerns about large variations in $M(\eta)$.

## Grassmannian parametrization

To demonstrate these shape representations, we use a data set containing 1,000 perturbations of 16 baseline airfoils from the NREL 5MW, DTU 10MW, and IEA 15MW reference wind turbines (Jonkman et al. 2009; Bak et al. 2013; Gaertner et al. 2020). The baseline airfoils are defined by the nominal 18 CST coefficients with the trailing edge thickness coefficients set to zero. We then perturb these 18 coefficients by up to 20% of their original value to create the data set.

Figure 3(a) shows a marginal 2D slice through the 18-dimensional space of CST coefficients defining the collection of shapes under consideration. Note that across the 16 baseline shapes, the groups of perturbations to nominal CST coefficients create a complex, highly disjoint design domain. This can significantly impact the performance of various AI/ML algorithms to analyze airfoils across this domain. We next demonstrate how the proposed representation addresses these issues with CST parametrization.

---

[4]A geodesic $\tilde{\gamma}_{k,k+1}$ is the shortest path between two points of a manifold and represents a generalized notion of the "straight line" in this non-linear topology.

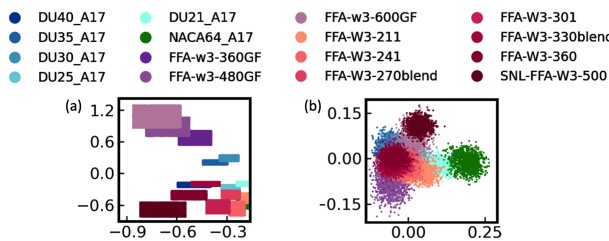

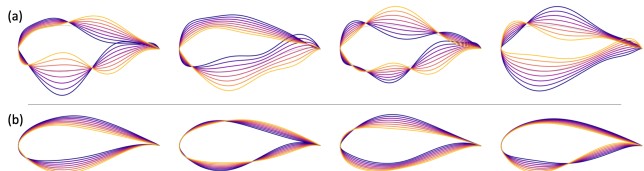

Figure 3: Comparison of the airfoil data over (a) 2 of the 18 total CST parameters and (b) 2 of the 4 total normal coordinates with colors indicating different classes of airfoils.

## Principal geodesic deformations

To infer a parametrized design space of airfoils over the Grassmannian, we use Principal Geodesic Analysis (PGA) (Fletcher, Lu, and Joshi 2003), a generalization of Principal Component Analysis (PCA) over Riemannian manifolds. PGA is a data-driven approach that determines principal components as elements in a *central tangent space*, $T_{[\tilde{\boldsymbol{X}}_0]}\mathcal{G}(n,2)$, given a data set represented as elements in a smooth manifold. In this way, PGA constitutes a manifold learning procedure for computing an important submanifold of $\mathcal{G}(n,2)$ representing a design space of physically relevant airfoil shapes inferred from provided data (Grey 2019).

First, we compute the Karcher mean $[\tilde{\boldsymbol{X}}_0]$ by minimizing the sum of squared (Riemannian) distances to all shapes in the data (Fletcher, Lu, and Joshi 2003). Second, we perform an eigendecomposition of the covariance of samples in the image of the Riemannian inverse exponential, $\text{Log}_{[\tilde{\boldsymbol{X}}_0]}:$ $\mathcal{G}(n,2) \rightarrow T_{[\tilde{\boldsymbol{X}}_0]}\mathcal{G}(n,2)$. This provides principal components as a new basis for a subspace of the tangent space. Finally, we map LA-standardized airfoils to normal coordinates of the tangent space at the Karcher mean via inner products with the computed basis—where $[\tilde{\boldsymbol{X}}_0]$ corresponds to the origin in normal coordinates, analogous to centering the data.

Based on the strength of the decay in eigenvalues, we take the first $r$ eigenvectors as a reduced basis for PGA deformations. Specifically, at a central airfoil $[\tilde{\boldsymbol{X}}_0]$ (e.g., Karcher mean), PGA results in an $r$-dimensional subspace of the tangent space, denoted $\text{span}(\boldsymbol{U}_r) \subseteq T_{[\tilde{\boldsymbol{X}}_0]}\mathcal{G}(n,2)$. We define normal coordinates $\boldsymbol{t} \in \mathcal{U} \subset \mathbb{R}^r$ where compact $\mathcal{U}$ contains the PGA data with appropriate distribution, e.g., uniform over an ellipsoid containing the data. Then, the set of all linear combinations of the principal components $\boldsymbol{U}_r\boldsymbol{t}$ defines an $r$-dimensional domain over $T_{[\tilde{\boldsymbol{X}}_0]}\mathcal{G}(n,2)$. This parametrizes a section of the Grassmannian ($r$-submanfiold) given by the image of the Riemannian exponential map, for all $\boldsymbol{t} \in \mathcal{U} \subset \mathbb{R}^r$,

$$\mathcal{A}_r = \left\{ [\tilde{\boldsymbol{X}}] \in \mathcal{G}(n,2) : [\tilde{\boldsymbol{X}}] = \text{Exp}_{[\tilde{\boldsymbol{X}}_0]}(\boldsymbol{U}_r\boldsymbol{t}) \right\}. \quad (2)$$

Truncating the principal basis to the first $r = 4$ components (based on the rapid decay in PGA eigenvalues), we significantly reduce the number of parameters needed to define a perturbation to an airfoil. Consequently, we have

Figure 4: A series of random corner-to-corner sweeps through (a) the CST and (b) principal geodesic design spaces partially visualized in Figure 3.

"learned" a 4-dimensional data-driven manifold of airfoils, $\mathcal{A}_4$, which are independent of affine deformations. New parameters are now coordinates of this four-dimensional subspace $\boldsymbol{t} \in T_0\mathcal{A}_4 \cong \mathbb{R}^4$ over the tangent space at the Karcher mean (our analogous origin for $\mathcal{A}_r$).

Figure 3(b) shows a 2D marginal slice of the airfoil data projected onto the four-dimensional PGA basis—i.e., a discrete distribution of $\boldsymbol{t} \in T_0\mathcal{A}_4$. Note that this design space roughly resembles a mixture of overlapping Gaussian distributions across the diverse family of airfoils. Compared to the CST representation, such a design space is significantly easier to infer or represent in the context of AI and ML algorithms. Further, extrapolation to shapes beyond the point cloud is significantly less volatile in this framework compared to CST. Figure 4 shows four random corner-to-corner sweeps (defined by bounding hyperrectangles) through the CST and principal geodesic design spaces. In CST space, it is difficult to define a single design space that covers the range of airfoils under consideration while allowing for smooth deformations between them. Conversely, all shapes generated using the proposed Grassmannian methodology result in valid airfoil designs while creating a rich design space worth investigation.

## Consistent blade deformations

Blade perturbations are constructed from deformations to each of the given cross-sectional airfoils in *consistent directions* over $\boldsymbol{t} \in T_0\mathcal{A}_4$. Since a perturbation direction is defined in the tangent space of Karcher mean, we utilize an isometry (preserving inner products) called parallel transport to smoothly "translate" the perturbing vector field along separate geodesics connecting the Karcher mean to each of the individual ordered airfoils. The result is a set of consistent directions (equal inner products and consequently equivalent normal coordinates in the central tangent space) over ordered tangent spaces $T_{[\tilde{\boldsymbol{X}}_k]}\mathcal{G}(n,2)$ centered on each of the nominal $[\tilde{\boldsymbol{X}}_k]$ defining the blade. An example of consistently perturbed sequence of cross-sectional airfoils is shown in Figure 2. Finally, these four principal components are combined with three to six independent affine parameters constituting a full set of 7-10 parameters describing a rich feature space of 3D blade perturbations.

The benefits of coherent shape deformations coupled with a natural framework for interpolating 2D shapes into 3D blades and the decoupling of affine and higher-order deformations make Grassmann-based shape representation a powerful tool enabling AI/ML-driven aerodynamic design.

## Acknowledgements

This work was authored in part by the National Renewable Energy Laboratory, operated by Alliance for Sustainable Energy, LLC, for the U.S. Department of Energy (DOE) under Contract No. DE-AC36-08GO28308. Funding partially provided by the Advanced Research Projects Agency-Energy (ARPA-E) Design Intelligence Fostering Formidable Energy Reduction and Enabling Novel Totally Impactful Advanced Technology Enhancements (DIFFERENTIATE) program. The views expressed in the article do not necessarily represent the views of the DOE or the U.S. Government. This work is U.S. Government work and not protected by U.S. copyright. A portion of this research was performed using computational resources sponsored by the Department of Energy's Office of Energy Efficiency and Renewable Energy and located at the National Renewable Energy Laboratory.

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
