# OpenReview forum: "Grassmannian Shape Representations for Aerodynamic Applications"
_AAAI.org/2022/Workshop/ADAM — AAAI 2022 Workshop ADAM_

### Official Review · Reviewer_rmjv · 2021-11-26
**Review for the novel Grassmannian Shape Representations of Aerodynamic Applications**

**Rating:** 6
**Confidence:** 3

**Review:**

Summary: this paper presents a novel airfoil shape design method based on data-driven techniques. Specifically, the authors proposed a novel representation of shapes which decouples affine-style deformations from a rich set of data-driven deformations over a submanifold of the Grassmannian. By comparing the principal geodesic deformations and consistent blade deformations with the traditional affine deformations, data-driven techniques show the critical efforts on representations and parameterizations of airfoil design problems in aerospace domain. This proposed method is very interesting to apply data-driven techniques for complex airfoil shape design. This work can provide some useful insights to the aerospace area on the airfoil shape design. The work is easy to follow and well written. While I have a couple concerns in mind. First, though data-driven techniques may provide better solutions, how practical is it when these are applied to real design problems? Since affine transformations are simple and easy to be implemented in practice. The author should discuss the difference in the paper. Second, how complex is the Grassmannian shape representations in theory? Though the author have shown clearly the derivation in the work, it would be better to see some complexity analysis in the paper.

---

### Official Review · Reviewer_MERY · 2021-12-01
**Interesting work**

**Rating:** 7
**Confidence:** 3

**Review:**

The paper proposes a new Grassmanian manifold based shape representation for airfoils. The authors suggest that such a representation may be more conducive to AI/ML algorithms. They also show the advantages of Grassmanian representations over CST, the current state of art representations for airfoils.

Pros:
1. The proposed representation is intuitive and well motivated for airfoil shape representation.
2. Fig.3b shows that the compressed representation shows that Grassmanian representations are similar to a mixture of Gaussians whereas the prevailing CST representations are more discrete spaces. This shows some advantages of the representations for gradient based optimization.
3. The paper is well written and presents useful comparisons to illustrate the advantage of the proposed model.

Cons:
1. The paper could do with a clear application of an AI/ML task to evidence the claims.

In summary, the paper appears to be a good addition to the workshop and will be of interest to the community.